# Unpacking the asymmetric impact of exchange rate volatility on trade flows: A study of selected developed and developing Asian economies

Umar Nawaz Kayani[1], Ahmet Faruk Aysan[2]\*, Azeem Gul[3], Syed Arslan Haider[4], Sareer Ahmad[5]

1 College of Business, Al Ain University, Abu Dhabi, UAE, 2 Qatar Foundation, College of Islamic Studies, Hamad Bin Khalifa University, Doha, Qatar, 3 Department of International Relations, National University of Modern Languages, Islamabad, Pakistan, 4 Department of Management, Sunway Business School (SBS), Sunway University, Petaling Jaya, Selangor Darul Ehsan, Malaysia, 5 School of Economics, Quaid-I-Azam University, Islamabad, Pakistan

\* aaysan@hbku.edu.qa

**Data Availability Statement:** Quarterly data spanning from 1980I to 2020IV were collected mainly from International Financial Statistics (IFS) under the IMF. Detailed information regarding the

## Abstract

Maintaining a stable exchange rate is a challenging task for the world, especially for developing economies. This study examines the impact of asymmetric exchange rates on trade flows in selected Asian countries and finds that the effects of increased exchange rate volatility on exports and imports differ among Pakistan, Malaysia, Japan, and Korea. The quarterly data from the period 1980 to 2018 is collected from the International Financial Statistics (IFS) database maintained by the International Monetary Fund (IMF). We employ both linear and non-linear Autoregressive Distributed Lag (ARDL) models for estimation. The non-linear models yielded more significant findings, while the linear models did not indicate any significant effects of exchange rate volatility on trade flows. The results of the study suggest that in the case of Pakistan, both the linear and non-linear models indicate that increased exchange rate volatility adversely affects exports and imports, while decreased volatility enhances both. This implies that stabilizing the exchange rate would be beneficial for Pakistan's trade. In contrast, the linear model applied to Malaysia shows no long-run effects of exchange rate volatility on exports. However, the result suggests that decreased volatility stimulates Malaysia's exports. Therefore, in the case of Malaysia, stabilizing the exchange rate could contribute to boosting exports. We also found that increased exchange rate volatility boosts exports of Japan. On the other hand, decreased volatility hurts exports of Japan. As for the long-run effects of exchange rate volatility on imports, we found that increased volatility boosts imports of Korea. The study provides various policy implications regarding the impact of exchange rate volatility on trade flows in developing economies. The study highlights the importance of country-specific considerations in understanding the impact of exchange rate volatility on trade flows, and has important policy implications for promoting trade and economic growth in these nations. It emphasizes the need to model exchange rate volatility

variables used in the study, including their definition, description, and source, are provided in Appendix A. The link for accessing data is as below: https://data.imf.org/?sk=388DFA60-1D26-4ADE-B505-A05A558D9A42&sId=1479329132316

**Funding:** The funders had no role in study design, data collection and analysis, decision to publish, or preparation of the manuscript.

**Competing interests:** he authors have declared that no competing interests exist.

separately for developed and developing countries and to continue research and analysis to identify ways to mitigate its negative effects on the economy.

## 1. Introduction

Exchange rates play a significant role in the economies of both developing and developed nations worldwide. The process of globalization has also reduced the importance of national boundaries, leading to increased economic integration and international trade [1, 2]. Exchange rates facilitate international trade by making imports and exports easier, while also serving as a means of transmitting funds between nations. However, fluctuations in exchange rates can create uncertainty in foreign trade, leading to speculation and disruptions in resource allocation, prices, and output levels [3]. To maintain an efficient exchange rate, the central bank of a nation intervenes in the foreign exchange market by injecting foreign exchange reserves when the exchange rate goes against the country's interest [4]. Conversely, when the exchange rate is in favor of a country, the central bank purchases foreign exchange, thereby maintaining stability in the exchange rate. Exchange rate variability is mainly caused by exchange rate shocks, domestic and foreign interest rate shocks, and terms of trade shocks [5]. Bahmani-Oskooee and Saha [6] investigates the impact of exchange rate volatility on the trade flow of India and its major trading partners. The evidence shows that short-run asymmetry exists in almost all industries while long-run asymmetry exists in half of the sample. The India largest trading partner China with trade share 11.17 percent of trade share has a significant positive impact on India's exports but a decline in volatility has no impact. Similarly, the second largest trading partner US shows that an increase rupee-dollar volatility has a positive long run effect on both Indian export and Import from US but decrease the volatility have no impact. Iqbal et al. [7] explores the impact of asymmetric exchange rate on Indian economic growth using the Hedrick Prescott Filter method. In asymmetric scenario the exchange rate misalignment has a negative impact on the Indian economic growth. However, after employing the NARDL, the evidence in favor of asymmetric effect is observed. Iqbal et al. [7] investigate the impact of exchange rate volatility and India-US commodity trade. According to the findings, rupee-dollar volatility has a short-term impact on 25 importing industries and a long-term impact on 15 sectors. Similarly, the third country effect reveals that Rupee-Yen volatility has a short and long-term impact on 9 Indian importing businesses.

The fluctuations in the exchange rate of the US dollar have garnered considerable attention in academic circles since the breakdown of the Bretton Woods system. Uncertainty in exchange rates has significant implications for the overall economic climate. The exchange rate (ER) directly affects international commerce, and fluctuations in its value can have a notable impact on the volume and value of international trade [8]. As such, the exchange rate is a crucial factor that economists and policymakers alike consider. For instance, countries that anticipate an increase in the exchange rate may prompt their governments and major banks to continually update their predicted inflation rates. The resulting volatility in exchange rates has been shown to have far-reaching consequences [9]. Bahmani-Oskooee et al. [10] did work on the Korean trade in ten service industries. They found that almost all ten service industries have short-run implications of imports and exports. In little more than half of industries, short-run effects convert into long-run effects. When we compared real trade in services to nominal commerce, the results did not differ much. Our findings were particular to the industry.

Free trade agreements have long been a subject of controversy, particularly concerning the impact on small and industrialized economies. While free trade is considered beneficial for small economies, it is also perceived to negatively impact industrialized economies by creating a competitive environment [11]. Nonetheless, free trade alliances have been found to have a positive impact on international trade. Bahmani-Oskooee et al. [10] investigate the impact of exchange rate on trade flow in case of G7 countries. In the short run, volatility affects almost all countries' trade flows asymmetrically. In the long run, while increasing volatility benefits French and Italian exports, it hurts German exports. Reduced volatility, on the other hand, affects French and Italian exports. In terms of G7 imports, increasing exchange rate volatility affects Canada, Germany, France, Italy, and the United Kingdom in the long run, while decreased volatility improves their imports.

Recent research by Bahmani-Oskooee and Saha [6] focuses on the impact of exchange rate volatility on Indian exports and imports with its fourteen largest trading partners. The study found that China, being the largest trading partner with a trade share of 11.17%, experienced a positive effect on Indian exports in response to an increase in real rupee-yuan exchange rate volatility, while a decrease in volatility had no significant impact. Similarly, the study revealed that the United States, as India's second-largest trading partner with a trade share of 10.48%, benefited from an increase in real rupee-dollar volatility in the long run, leading to favorable outcomes on both India's export and imports from the United States. However, a decrease in exchange rate volatility did not significantly affect India's trade with the United States. Wang [12] investigates the impact of asymmetric exchange rates on bilateral trade flow between the US and India. The findings imply that the effects of bilateral exchange rates on the US trade imbalance with China varied across exchange rate systems. There is an unequal long-run effect of bilateral exchange rates. The findings also show that the weakening of the Chinese currency will have little effect on US trade imbalances with China [2].

The impact of the exchange rate on export and economic growth is a topic of ongoing debate among researchers, with a lack of consensus in the current empirical literature regarding the magnitude and direction of the potential impact [13]. However, it is widely accepted that the exchange rate can affect domestic employment and production in a variety of ways, including through export-led growth. A competitive exchange rate can make imports more expensive and exports cheaper, potentially correcting balance of payment deficits and improving domestic output and employment if the Marshall-Lerner hypothesis holds true. However, this condition is not necessarily fulfilled in every country, particularly in the era of globalization and vertically integrated industries. Bahmani-Oskooee et al. [6] investigates the asymmetric impact of exchange rate on trade flow between Pakistan and China. Finding reveal short-run asymmetric impacts of exchange rate volatility in almost all industries, with long-run asymmetric effects in 40–50 percent of them. Non-linear models produced more significant volatility effects than typical linear models.

Exchange rate fluctuations also have significant implications for investment flows and foreign debt servicing. In addition, in the short run, exchange rate depreciation can worsen the trade balance, while in the long run, it can improve it, a phenomenon known as the J-curve. Hafeez and Haseeb [14] concluded that trade openness and interest rate have a positive and significant impact on the economy, particularly in the absence of frequent structural shocks and political changes. Mahmood et al. [15] found a negative relationship between the volatility of exchange rate and foreign direct investment (FDI), while Hanif et al. [16] reported mixed results in their analysis of the response of exchange rate on macroeconomic variables.

Other studies have explored the relationship between inflation, gross domestic product (GDP) growth, and interest rate with exchange rate for both developed and developing countries. Semuel et al. [17] found that inflation rate, industrial production, and balance of payment

(BOP) have an association with exchange rate, while Ahmad et al. [18] reported a positive and significant relationship between exchange rate and balance of payment in Pakistan. Stable exchange rate, therefore, may establish a positive environment for empowering investment and improving balance of payment. Chu et al. [19] studies the statistical analysis of exchange rate of Bitcoin. They conducted a statistical analysis on the log-returns of Bitcoin's exchange rate against the United States Dollar, using fifteen popular parametric distributions in finance. Among them, the generalized hyperbolic distribution demonstrated the best fit. Chen [20] studies the impact of exchange rate volatility on trade and financial expansion. The finding shows that financial openness reported positive while trade openness has a negative impact on exchange rate volatility.

Chaudhary et al. [21] evaluated the response of exchange rate volatility on FDI in Asian countries and reported mixed results, with some countries exhibiting a significant association while others did not. Oladipupo et al. [22] empirically studied the response of exchange rate on balance of payment for Nigeria and reported a positive impact of exchange rate on the balance of payment, particularly if fiscal discipline is imposed.

In the context of South Asia, Pakistan is an emerging economy that heavily depends on oil imports and imports of modern technology and inputs for home production and consumption. The country has been facing a persistent trade deficit and low foreign reserves, with exports staying below US$25 billion and imports exceeding US$50 billion for the most recent decade, putting immense pressure on the external balance (Economic Survey of Pakistan, 2015–16). This poor performance can be attributed to the limited export basket, with Pakistani exports being highly concentrated in a few products, such as cotton, synthetic textiles, rice, and sports products. The Pakistani economy needs to diversify its exports to include high technology, high-value items to improve its external balance.

A study conducted by Aman et al. [23] utilized synchronous condition models, 2SLS and 3SLS, to investigate the impact of exchange rate on economic growth, highlighting significant effects through investment, FDI, and exports channels. Ramasamy and Abar [24] identified macroeconomic indicators such as interest rate, GDP growth, inflation, political strength, industrial production, Treasury bill, and unemployment as factors affecting exchange rate. Bhagwati and Krueger's [25] perspective emphasizes that reducing the anti-export bias through policy can lead to trade liberalization and decrease the import permit premium. Bosworth's [26] findings indicate a weak relationship between interest rate and economic growth. Additionally, Alam and Qazi's [27] research highlights the adverse impact of volatile effective exchange rate on Pakistan's aggregate exports, with evidence of co-integration.

To evaluate the response of exchange rate on trade flow using the Linear and nonlinear ARDL model. The findings of the Linear model depict that there are no significant effects of exchange rate volatility on trade flows, while the results of nonlinear models show relatively more significant outcomes. The result indicates that both linear and nonlinear model indicates that decreased exchange rate volatility will boost Pakistani exports and imports while increased exchange rate volatility will hurt both. Nevertheless, the linear model indicates that EXR volatility has no long run effects on Malaysia exports. The findings shows that decreased volatility of exchange rate has stimulate the exports of Malaysia, but increased exchange rate volatility has no effect. Consequently, Malaysia boosts its exports by stabilizing its exchange rate. We found additionally that increased exchange rate volatility boosts exports of Japan. On the other hand, decreased volatility hurts exports of Japan. As for the long-run effects of exchange rate volatility on imports, we found that increased volatility boosts imports of Korea.

The objectives of this study are to evaluate the response of exchange rate on export and import in selected Asian developing and developed countries. This study examines the impact of asymmetric exchange rates on trade flows in Pakistan, Malaysia, Japan, and Korea,

representing a mix of developing and developed countries in Asia. The selection of these countries is justified as it enables a comprehensive understanding of the effects of exchange rate volatility on trade flows across diverse economic contexts. With their significant economic importance and varying trade volumes, these countries offer valuable insights into the country-specific considerations involved. This understanding is crucial for formulating effective policies to promote trade and economic growth in each nation. Importantly, the study highlights the need to separately model exchange rate volatility for developed and developing countries. It emphasizes the importance of ongoing research to identify strategies that can mitigate the negative effects of exchange rate volatility on the economy. By focusing on these countries, the study contributes to a more coherent understanding of the complex relationship between exchange rate volatility, trade flows, and economic development.

The structure of the paper is as follows. The subsequent sections describe the model and methodology (section 2), report the results (section 3), and provide conclusions and recommendations (section 4).

## 2. Literature review

The exchange rate is a crucial component of international trade and a significant macroeconomic variable. Exchange rate fluctuations play a pivotal role in determining the trade balance, but high volatility in exchange rates can slow down the trade process. Handoyo et al., [28] investigate the impact of asymmetric exchange rates on manufacturing commodities exports In ASEAN-5 countries using NARDL model. The ARDL technique found that volatility had a considerable short term impact on 13 commodity exports. The Nonlinear ARDL method found that volatility has an impact on 19 commodity exports. The ARDL technique found that volatility had a considerable short term impact on 13 commodity exports. The Nonlinear ARDL method found that volatility has an impact on 19 commodity exports. Furthermore, findings from ARDL and Nonlinear ARDL imply risk-averse behavior by exporters in the long run. However, the nonlinear model indicates that volatility has an unbalanced influence on nearly all commodity exports in the long run. Numerous studies have been conducted to examine the correlation between exchange rate uncertainty and trade, yielding varied outcomes. Some researchers, such as Cushman [29], Akhtar and Hilton [30], Persson and Svensson [31], and Peree and Steinherr [32], have found that exchange rate fluctuations negatively affect trade volume. In contrast, Rogoff [33] discovered that exchange rate volatility poses significant challenges for exporters and importers. Arize [34] observed a negative and significant long-term relationship. Lanyi and Suss [35] reported that exchange rate volatility impacts the prices of domestic currency, exports, and imports, thereby hindering international transactions. It is widely believed that excessive exchange rate fluctuations have adverse effects on welfare by reducing international trade, impeding economic growth, and influencing investment decisions [36, 37]. Moreover, currency appreciation can have a detrimental impact on international trade.

Doğanlar [38] argued that if the model is correctly specified, the relationship between exchange rate instability and trade should not be negative; in fact, it should be theoretically positive. Abdoh et al. [39] concluded that interest rates and inflation do not significantly affect exchange rates, while exports have a positive impact on the exchange rate. Dimpfl and Schmidt [40] found no evidence of an exchange rate effect on Chinese exports, but negative reactions on the trade balance when dealing with Western trading partners such as the USA. Ozturk [41] examined the impact of exchange rate volatility on trade and found that increased volatility adversely affected international trade. In a study conducted by Abbas et al. [42] to examine the relationship between real interest rate, GDP, inflation, and exchange rate, significant

associations were found between GDP growth and the exchange rate, while no association was observed between interest rate, inflation, and the exchange rate. Rangrajan [43] argued that exchange rate instability could hinder the smooth functioning of international trade and the global economy, leading to higher prices for internationally traded goods as traders and banks add a risk premium to compensate for unexpected exchange rate fluctuations.

While previous studies have explored the impact of exchange rate volatility on international trade, this current study contributes to the literature by empirically investigating the asymmetric effects of exchange rate volatility on exports and imports in selected Asian economies, namely Pakistan, Malaysia, Korea, and Japan. The inclusion of both advanced and developing economies, as well as the consideration of trade agreements in the studied economies, adds to the significance of this research. The asymmetric approach is adopted to account for the different expectations traders may have during periods of increased and decreased exchange rate volatility.

However, despite the existing body of research on the relationship between exchange rate uncertainty and trade, there is a notable gap in the literature when it comes to investigating the specific asymmetric effects of exchange rate volatility on exports and imports in selected Asian economies, namely Pakistan, Malaysia, Korea, and Japan. While previous studies have examined the overall impact of exchange rate volatility on international trade, few have delved into the distinct effects on exports and imports separately. Moreover, the inclusion of both advanced and developing economies, as well as the consideration of trade agreements within the studied economies, further highlights the significance of addressing this research gap. By adopting an asymmetric approach that accounts for the differing expectations of traders during periods of increased and decreased exchange rate volatility, this study aims to contribute valuable insights to the existing literature.

## 3. Data and methodology

### 3.1. Data

The aim of this study is to analyze the relationship between exchange rates and trade flows in several Asian countries including Pakistan, Malaysia, Japan, and Korea. Quarterly data spanning from 1980I to 2020IV were collected mainly from International Financial Statistics (IFS) under the IMF. Detailed information regarding the variables used in the study, including their definition, description, and source, are provided in S1 Appendix.

### 3.2. Model specification

The study utilized econometric models to analyze the asymmetric impact of exchange rate on exports and imports in Pakistan, Korea, Japan, and Malaysia. The main determinants of trade flows, including scale variables such as REER and exchange rate volatility, were included in the models. The following export and import demand models were used based on previous research studies.

$$LnX_t = \beta + \beta_1 LnYW_t + \beta_2 LnREX_t + \beta_3 LnV_t + \mu_t \tag{A}$$

$$LnM_t = \alpha + \alpha_1 LnY_t + \alpha_2 LnREX_t + \alpha_3 LnV_t + \mu_t \tag{B}$$

Eq (A) demonstrates that the variable X represents exports to foreign countries. X is defined as the unit value of exports deflated by nominal exports and is presumed to be influenced by global income (YW), real exchange rate (REX), and real effective exchange rate volatility (REEV). However, this relationship may lead to asymmetric effects since any measurement of

volatility is based on the real exchange rate, which encompasses nominal exchange rate and prices. As Asymmetry we decompose the exchange rate into positive and negative components e-i., appreciation and depreciation and then evaluate its impact on trade flows. If the appreciation and depreciation impact on trade flow is reported differently it means that there is asymmetry while if the impact is same, then it mean that the impact of exchange rate on trade balance is symmetric manner.

To quantify exchange rate volatility, an econometric technique called Generalized Autoregressive Conditional Heteroskedasticity (GARCH) is employed. The choice of the GARCH model for volatility measurement in this study is justified by several factors. Firstly, the GARCH model is widely recognized and extensively used in financial econometrics for modeling and forecasting volatility. It has been proven to be effective in capturing the time-varying nature of volatility, which is a crucial aspect when analyzing exchange rate fluctuations. The GARCH model incorporates the auto-regressive and moving average components, allowing it to capture both the short-term and long-term dynamics of volatility.

Secondly, the GARCH model is particularly suitable for capturing asymmetric effects of volatility. It takes into account the phenomenon of volatility clustering, where periods of high volatility tend to be followed by more periods of high volatility, while periods of low volatility tend to be followed by more periods of low volatility. This asymmetric behavior is often observed in financial markets and is highly relevant when studying the impact of exchange rate volatility on trade flows. By using the GARCH model, this study aims to capture and quantify the asymmetric effects of exchange rate volatility on exports and imports separately, providing valuable insights into the differential impacts on trade flows in the selected Asian countries.

Additionally, the GARCH model allows for the consideration of country-specific factors and characteristics. This is crucial when analyzing the impact of exchange rate volatility on trade flows in different economies, as the effects can vary depending on various factors such as economic development, trade agreements, and policy frameworks. By employing the GARCH model, this study can consider the specific circumstances of Pakistan, Malaysia, Japan, and Korea, providing a more nuanced understanding of the relationship between exchange rate volatility and trade flows in these countries.

The country's own income (Y) and the real exchange rate (REX) along with the volatility of the effective exchange rate are believed to be responsible for a country's imports (M). The variable M represents total imports and is defined as the unit value of imports deflated by nominal imports. We expect the estimate of $\beta_1$ to be positive because imports tend to increase as the country's income rises. As the effective real exchange rate declines, we expect an estimate of $\alpha_1$ to be negative. This is because exports become more affordable in the foreign currency, whereas import costs rise due to the devaluation of the national currency.

Moreover, the erratic nature of the exchange rate may impact exports and imports in opposite directions. The resulting effects could be either positive or negative, depending on the estimation method employed. To obtain long-term forecasts of imports and exports, Eqs (A) and (B) must be estimated using any approach. Error correction models (ECM) can be used to determine the short-term impacts of all exogenous factors on exports and imports. Both short-term and long-term forecasts can be produced using Pesaran et al. [44] new technique in a single step.

$$\Delta LnX_t = \beta_1 + \sum_{j=1}^{n1}\beta_2 j\Delta LnX_{t-j} + \sum_{j=0}^{n2}\beta_3 j\Delta LnYW_{t-j} + \sum_{j=0}^{n3}\beta_4\Delta LnREX_{t-j} + \sum_{j=0}^{n4}\beta_5\Delta LnV_{t-1} \quad (C)$$
$$+\theta_1 LnX_{t-1} + \theta_2 LnYW_{t-1} + \theta_3 LnREX_{t-1} + \theta_4 LnV_{t-1} + \varepsilon_t$$

$$\Delta LnM_t = \alpha_1 + \sum_{j=1}^{n5}\alpha_2 j\Delta LnM_{t-j} + \sum_{j=0}^{n6}\alpha_3 j\Delta LnY_{t-j} + \sum_{j=0}^{n7}\alpha_4\Delta LnREX_{t-j} + \sum_{j=0}^{n8}\alpha_5\Delta LnV_{t-1} \quad \text{(D)}$$

$$+\rho_1 LnM_{t-1} + \rho_2 LnY_{t-1} + \rho_3 LnREX_{t-1} + \rho_4 LnV_{t-1} + \varepsilon_t$$

The coefficients assigned to the first-different variables in Eqs (C) and (D) have short-term and long-term effects, respectively. In order to align the long-run effects with models (A) and (B), the estimates of θ2-θ4 must be adjusted. The ECM models (C) and (D) suggest that trade flows respond symmetrically to changes in exogenous factors. However, our main aim is to examine the asymmetric effects of exchange rate volatility, and we follow Shin et al. [45] by using partial sums to distinguish between increasing and decreasing volatility.

$$POS_t = \sum_{j=1}^{t}\Delta LnV_j^+ = \sum_{j=1}^{t}MAX(\Delta LnV_{j,0})$$

$$NEG_t = \sum_{j=1}^{t}\Delta LnV_j^- = \sum_{j=1}^{t}MIN(\Delta LnV_{j,}) \quad \text{(E)}$$

Here POSt shows the positive change in LnV and the time series variable which shows volatility increased only. Similarly, NEGt is shows change negative in volatility and indicates that decreases volatility only. Now we are moving back to the Eqs (C) and (D) and replace LnVt by POSt and NEGt. The new result of ECM model as follows:

$$\Delta LnX_t = d_1 + \sum_{j=1}^{n1} d_2 j\Delta LnX_{t-j} + \sum_{j=0}^{n2} d_3 j\Delta LnYW_{t-j} + \sum_{j=0}^{n3} d_4 j\Delta LnREX_{t-j} + \sum_{j=0}^{n4} d_5 j\Delta POS_{t-j} +$$

$$\sum_{j=0}^{n5} d_6 j\Delta NEG_{t-j} + \lambda_1 LnX_{t-1} + \lambda_2 LnYW_{t-1} + \lambda_3 LnREX_{t-1} + \lambda_4 POS_{t-1} + \lambda_5 NEG_{t-1} + \mu \quad \text{(F)}$$

$$\Delta LnM_t = e_1 + \sum_{j=1}^{n6} e_2 j\Delta LnM_{t-j} + \sum_{j=0}^{n7} e_3 j\Delta LnY_{t-j} + \sum_{j=0}^{n8} e_4 j\Delta LnREX_{t-j} + \sum_{j=0}^{n9} e_5 j\Delta POS_{t-j} +$$

$$\sum_{j=0}^{n10} d_6 j\Delta NEG_{t-j} + \pi_1 LnM_{t-1} + \pi_2 LnY_{t-1} + \pi_3 LnREX_{t-1} + \pi_4 POS_{t-1} + \pi_5 NEG_{t-1} + \mu_t \quad \text{(G)}$$

These models are nonlinear ARDL models and in the form of non-linearity created from the method creating partial sum variables. Shin et al. [45] express linear and nonlinear model both could be estimated by OLS (Ordinary least Square) Method.

## 4. Results and discussions

We have estimated the linear models (C) and (D), as well as the nonlinear models (F) and (G), for Pakistan, Malaysia, Japan, and Korea using quarterly data from the period 1980I to 2018IV. For each model, we considered a maximum of 8 lags and used AIC (Akaike's Information Criterion) to determine the optimal specification of the models, subject to critical values. We have also conducted diagonal tests to ensure the validity of our estimates. Starting with the export demand linear model (C) for each country, we present the results in Tables 1 and 2.

Based on the estimates of short-run volatility coefficients presented in Table 1, it can be observed that every country, except Malaysia, has at least one coefficient that is statistically significant. This means that there is evidence of a significant relationship between currency rate volatility and exports in these countries. Specifically, in the case of Pakistan, Korea, and Japan, the coefficient estimates indicate that currency rate volatility has substantial long-term impacts on exports. In Pakistan, the coefficient estimates suggest that increased volatility has a negative effect on exports. In contrast, both Korea and Japan experience both negative and positive long-term effects of currency rate volatility on their exports, indicating a more complex relationship. However, it should be noted that Malaysia does not show any statistically significant coefficients in the short run, implying that currency rate volatility does not have a significant

**Table 1. Short run coefficient estimates on volatility on exports.** Linear export demand model C.

| Country | Lag number on ΔLnV | | | | | | | | |
|---|---|---|---|---|---|---|---|---|---|
| | 0 | 1 | 2 | 3 | 4 | 5 | 6 | 7 | 8 |
| **Pakistan** | -1.45 (1.78)* | 12.5 (1.69)* | 18.2 (1.04) | 16.13 (0.01) | 29.07 (0.15) | 5.227 (0.76) | 3.47 (0.84) | 5.84 (0.03) | |
| **Malaysia** | 1.94 (0.22) | 7.06 (0.56) | 1.92 (0.78) | 8.91 (1.38) | -11.31 (0.92) | | | | |
| **Korea** | 10.19 (0.95) | -13.35 (1.21) | -9.73 (1.67)* | -12.86 (1.08) | | | | | |
| **Japan** | 6.46 (1.22) | -0.86 (0.43) | -13.20 (2.24)** | -3.74 (0.34) | -5.63 (1.20) | -4.50 (1.06) | | | |

Notes: The critical Values of t test are 1.64 & 1.96 at 10% and 5% significance level, *and ** indicates significance level at 10% and 5%.

impact on exports in the short term in Malaysia. Overall, the paragraph highlights that the impact of currency rate volatility on exports varies among the selected countries. Pakistan sees a consistently negative effect, while Korea and Japan experience a mix of negative and positive effects. The absence of significant coefficients for Malaysia suggests a lack of short-term impact. These findings emphasize the importance of considering country-specific factors and dynamics when examining the relationship between currency rate volatility and exports.

We ensured that residuals were not auto-correlation-free by using a variety of diagnostic tests in Table 2. We've also employed the LM test, which is distributed at $\chi^2$ with one degree of freedom since we're looking for first-order auto-correlations. A look at the models reveals that it has no bearing on them at all. Misspecification may also be detected using the RESET test, which has $\chi^2$ with one degree of freedom and is inconsequential for all except China. Finally, we looked the goodness of fit using an updated R2. As a result, all of the models have an excellent fit. The ECM associated with a linear import model and estimates in Table 3 will be calculated next Table 4.

Based on the information provided in Table 3, it is evident that currency rate volatility has a significant short-term impact on the imports of all the nations under consideration. This implies that fluctuations in exchange rates have an immediate effect on the imports of these countries.

Furthermore, the paragraph suggests that the short-term actions related to currency rate volatility in Pakistan and Malaysia may have long-term consequences. This indicates that the effects of short-term fluctuations in exchange rates can persist and continue to influence imports over a longer period in these two countries. Additionally, Table 4 shows a significant coefficient of co-integration between Pakistan and Malaysia when measuring volatility. This implies that there is a strong long-term relationship between the two countries in terms of exchange rate fluctuations. Specifically, both Pakistan and Malaysia experience a negative impact on their imports because of exchange rate fluctuations.

**Table 2. Long run coefficient estimates of linear demand export model (C).**

| Country | Long run estimates Diagnostic test | | | | | | | | |
|---|---|---|---|---|---|---|---|---|---|
| | Constant | LnYW | LnREX | LnV | F stat | ECMt-1 | LM | RESET | ADJ.R2 |
| **Pakistan** | 13.31 (3.32)** | -1.04 (1.70)** | 2.11 (2.11)** | -11.01 (2.61)** | 3.89* | -0.18 (3.33)** | 1.9 | 1.71 | 0.99 |
| **Malaysia** | 1.02 (1.01) | -1.12 (1.78)* | -0.81 (0.13) | 0.12 (1.02) | 1.18 | -0.14 (2.18)** | 1.21 | 2.01 | 0.99 |
| **Korea** | -15.71 (1.12) | 1.34 (3.51)** | -4.04 (1.87)* | 14.61 (1.91)* | 4.11* | -0.13 (3.23)** | 1.09 | 1.77 | 0.97 |
| **Japan** | 12.33 (3.11)** | 0.13 (0.17) | -4.32 (2.42)** | 12.21 (2.92)** | 3.89* | -0.21 (4.61)** | 2.17 | 1.09 | 0.98 |

Note: the absolute value of the t ratios * and** indicates that coefficient estimates are at 10% and 5% level respectively. The F test critical value for co integration when there are 3 variables is 3.77 (4.35) at 10%(5%) level of significance, t test critical value for significance of ECMt-1 is -3.46(-3.78) at 10% (5%) level of significance when k = 3

**Table 3. Short run coefficient estimates of volatility on imports estimates linear import demand model (D)).**

| Country | Lags on $\Delta$Ln V | | | | | | | | |
|---|---|---|---|---|---|---|---|---|---|
| | 0 | 1 | 2 | 3 | 4 | 5 | 6 | 7 | 8 |
| **Pakistan** | 8.21 (1.06) | 18.07 (1.89)* | -12.97 (1.31) | 28.31 (2.09)** | | | | | |
| **Malaysia** | -18.65 (1.06) | 56.44 (2.32)** | 28.32 (1.90)* | 25.03 (1.54) | 46.64 (1.87)* | | | | |
| **Korea** | 52.21 (1.74)* | -24.92 (0.72) | 29.32 (0.89) | | | | | | |
| **Japan** | 6.32 (1.90)* | -1.10 (1.02) | -3.08 (1.01) | -7.08 (1.32) | -3.09 (1.05) | | | | |

Notes: * indicates significance at 10% level and ** at 5% level.

In summary, the paragraph highlights that currency rate volatility has a notable short-term impact on imports across all the nations considered. It also suggests that the short-term actions related to exchange rate fluctuations in Pakistan and Malaysia can have long-lasting effects. Furthermore, the significant coefficient of co-integration between Pakistan and Malaysia indicates a strong long-term relationship in terms of the impact of exchange rate fluctuations on imports, with both countries experiencing negative effects.

The paragraph explains the results obtained from various tables related to short-term and long-term impacts of exchange rate volatility on exports and imports in different nations. Table 5 presented that all nations, except Korea and Malaysia, have at least one significant coefficient in the short-run estimations. This indicates that exchange rate volatility has a short-term impact on exports across all nations, although the results suggest that the impacts vary between countries. The estimates related to positive exchange rate changes (ΔPOS) at lag order (t-j) are more accurate than the estimates related to negative exchange rate changes (ΔNEG). In the case of Pakistan and Japan, the short-term asymmetric effects become significant over the long term, but this is not observed in Korea. The remaining three nations (excluding Korea) also show significant coefficients in the positive (POS) or negative (NEG) variables, indicating that rising volatility negatively affects Pakistani exports while benefiting Japan's. Furthermore, the paragraph discusses the impacts of increased volatility and decreased volatility on exports in different nations. It states that increased volatility helps Pakistani exports and Malaysia's but weakens Japan's exports. On the other hand, lower exchange rate volatility boosts exports from Malaysia and Pakistan. The paragraph then shifts to the estimation of a nonlinear import demand model in Tables 6 and 7 for each nation. It mentions that significant coefficients are found in at least one of the positive or negative variables (ΔPOS or ΔNEG) for all countries except Japan in terms of short-term outcomes. The short-term impacts of these estimates differ from those linked to the lag order variable (ΔPOS or ΔNEG), as the two variables are asymmetrically related.

In the case of Pakistan and Korea, the data shows exclusively short-term commutative asymmetric effects. Table 8 indicates that these short-term asymmetric effects transform into

**Table 4. Long run coefficient estimates of linear import demand model (D).**

| Country | Constant | LnY | LnREX | LnV | F statistic |
|---|---|---|---|---|---|
| Pakistan | 9.11 (3.19)* | -4.12 (1.77)* | 3.33 (1.69)* | -8.18 (-1.78)* | 3.79* |
| Malaysia | 9.71 (1.71)* | -2.15 (2.87)** | 8.52 (1.78)* | -43.21 (2.82)** | 1.99 |
| Korea | -5.65 (1.63) | 1.16 (2.91)** | -1.41 (1.55) | 2.72 (1.52) | 2.01 |
| Japan | -3.65 (1.08) | -0.18 (1.95)* | -1.53 (0.33) | 3.35 (1.08) | (4.16)* |

Note: the absolute value of the t ratios * and** indicates that coefficient estimates are at 10% and 5% level respectively.

**Table 5. Short run coefficient estimates of ΔPOS and ΔNEG in nonlinear export demand model (F).**

| Country | Lags on ΔPOS | | | | | | | | |
|---|---|---|---|---|---|---|---|---|---|
| | 0 | 1 | 2 | 3 | 4 | 5 | 6 | 7 | 8 |
| Pakistan | 2.32 (1.02) | 13.34 (0.45) | 14.43 (2.43)** | 13.98 (3.23)** | 17.64 (1.99)** | -8.42 (1.13) | -3.24 (1.27) | -5.16 (1.60) | |
| Malaysia | -8.35 (0.43) | 14.33 (1.03) | 12.32 (1.41) | | | | | | |
| Korea | 19.43 (0.53) | 23.21 (0.46) | -2.64 (0.35) | 5.43 (0.11) | | | | | |
| Japan | 6.25 (1.78)* | -4.43 (0.78) | -11.54 (2.56)** | | | | | | |
| Country | Lags on ΔNEG | | | | | | | | |
| | 0 | 1 | 2 | 3 | 4 | 5 | 6 | 7 | 8 |
| Pakistan | 6.34 (0.49) | 12.92 (0.09) | 19.22 (2.23)** | 15.91 (3.34)** | 23.33 (1.92)* | -6.32 (1.30) | -3.32 (1.12) | | |
| Malaysia | -8.02 (0.68) | 15.09 (0.79) | 10.46 (0.29) | 19.31 (1.22) | 1.21 (0.12) | | | | |
| Korea | 11.23 (1.23) | 17.53 (0.97) | -4.03 (0.39) | 5.03 (1.19) | 17.03 (1.01) | | | | |
| Japan | 7.37 (1.77)* | | | | | | | | |

Notes: The critical Values of t test are 1.64 & 1.96 at 10% and 5% significance level, *,** indicates significance level at 10% and 5%.

substantial long-term impacts for Pakistan, Malaysia, and Korea when the positive or negative variables have significant coefficients. In particular, a more volatile currency rate harms imports from Pakistan and Malaysia but benefits Korean exports. Lower exchange rate volatility leads to increased exports from Malaysia and Pakistan. Volatility increases have long-term effects on Korea's imports, while volatility decreases do not. The paragraph acknowledges that there is no consensus in the literature regarding the positive or negative impact of exchange rate volatility on trade flows, and studies produce mixed results. However, economic theory suggests that when a country's exchange rate increases relative to other countries, its goods and services become more expensive, leading to a decrease in exports and an increase in imports. In summary, the paragraph provides a logical explanation of the results obtained from various tables, highlighting the short-term and long-term impacts of exchange rate volatility on exports and imports in different nations, as well as the asymmetric effects observed in certain cases. It also mentions the lack of consensus in the literature and emphasizes the theoretical relationship between exchange rate changes and trade flows.

## 5. Conclusion

The primary objective of this study was to investigate the effects of asymmetric exchange rates on trade flows in selected Asian countries, namely Pakistan, Malaysia, Japan, and Korea. To achieve this objective, the researcher utilized quarterly data from 1980 to 2018 obtained from the IMF's International Financial Statistics (IFS). Nonlinear model estimation was employed to account for the asymmetric nature of the relationship between exchange rate volatility and trade flows. The findings of the study indicate that the short-term effects of exchange rate

**Table 6. Long run coefficient estimates of the nonlinear export model (F).**

| Country | Constant | LnYW | LnREX | POS | NEG |
|---|---|---|---|---|---|
| Pakistan | -7.91 (1.91)* | -0.12 (1.51) | 3.91 (2.19)** | -7.98 (1.89)* | -8.75 (1.73)* |
| Korea | 11.71 (1.29) | -1.71 (1.26) | -1.71 (1.06) | 11.1 (1.22) | 2.22 (1.21) |
| Malaysia | -63.60 (2.15)** | -1.05 (2.59)** | 13.80 (1.87)* | -3.28 (1.08) | -10.72 (2.26)** |
| Japan | 25.42 (3.33)** | 0.33 (1.25) | -2.62 (2.62)** | 12.32 (2.92)** | 13.01 (2.82)** |

Note: the absolute value of the t ratios * and** indicates that coefficient estimates are at 10% and 5% level respectively.

**Table 7. Short run coefficient estimates of ΔPOS and ΔNEG in nonlinear import demand model (G).**

| Country | Lag number on ΔPOS | | | | | | | | |
|---|---|---|---|---|---|---|---|---|---|
| | 0 | 1 | 2 | 3 | 4 | 5 | 6 | 7 | 8 |
| Pakistan | 10.09 (2.06)** | 11.12 (2.22)** | 5.19 (1.23) | 17.01 (3.93)** | 19.14 (1.93)* | -10.81 (1.35)* | 17.71 (191)* | 10.01 (0.19) | |
| Malaysia | -2.33 (1.09) | 19.81 (2.12)** | 18.62 (1.01) | 6.91 (1.73)* | 15.32 (1.23) | 15.22 (1.24) | | | |
| Korea | 16.12 (2.04)** | -12.52 (1.411) | 9.51 (1.87)* | -10.51 (1.08) | 16.01 (1.51) | | | | |
| Japan | 3.63 (0.33) | -1.12 (1.12) | -2.72 (1.02) | -2.14 (1.54) | | | | | |
| Country | Lag number on ΔNEG | | | | | | | | |
| | 0 | 1 | 2 | 3 | 4 | 5 | 6 | 7 | 8 |
| Pakistan | 12.32 (1.91)* | 11.51 (3.12)** | 11.01 (2.12)** | 11.71 (3.11)** | 11.91 (1.89)* | -12.93 (0.89) | | | |
| Malaysia | -1.46 (1.61) | 9.11 (2.39)** | 13.02 (1.05) | 10.81 (91.85)* | 17.01 (1.50) | | | | |
| Korea | 11.11 (1.72)* | -12.81 (1.08) | 11.46 (1.78)* | | | | | | |
| Japan | 4.62 (1.29) | | | | | | | | |

Notes: The critical Values of t test are 1.64 & 1.96 at 10% and 5% significance level, *, ** indicates significance level at 10% and 5%.

volatility on exports and imports were observed in all countries, as revealed by both linear and nonlinear models. However, the long-term impacts varied across the four countries, with the nonlinear model providing more significant estimates. Specifically, the study revealed that increased exchange rate volatility had a negative impact on Pakistani exports but a positive impact on Japanese exports. The volatility of the Pakistani currency played a crucial role in driving Pakistani exports, while in Japan, increased volatility positively influenced export levels. Regarding imports, higher exchange rate volatility had a long-term negative impact on Pakistani and Malaysian imports, while Korean imports were positively influenced by increased volatility. Conversely, a decrease in exchange rate volatility encouraged imports from Malaysia and Pakistan. The study highlights the importance of country-specific considerations in understanding the impact of exchange rate volatility on trade flows. The findings have important policy implications, providing guidance for policymakers aiming to promote trade and economic growth in these nations. Additionally, the study examined the impact of trade rules and regulations, such as tariffs and coherence levels, on the outcomes in each country. The varying trade rules and regulations among the countries significantly influenced the observed outcomes. In conclusion, this study contributes to the understanding of exchange rate volatility and its impact on trade flows, economic growth, and investment decisions. It emphasizes the need to model exchange rate volatility separately for developed and developing countries and highlights the importance of accurate forecasts to comprehend its effects on the economy. By identifying country-specific factors and potential remedies, policymakers can effectively address the negative impacts of exchange rate volatility and foster sustainable economic development.

Based on the results of the study, several policy-relevant suggestions can be made:

**Table 8. Long run coefficient estimates of the nonlinear import demand model (G).**

| Country | Constant | LnY | LnREX | POS | NEG |
|---|---|---|---|---|---|
| Pakistan | -4.62 (1.49) | -2.01 (1.01) | 2.09 (2.23)** | -5.62 (1.90)* | -4.74 (1.87)* |
| Korea | 5.55 (1.52) | 1.21 (2.82)** | -3.22 (1.23) | 5.45 (1.67)* | 8.32 (1.16) |
| Malaysia | -14.93 (1.87)* | -8.16 (1.88)* | 12.58 (1.77)* | -12.27 (1.96)** | -12.62 (2.35)** |
| Japan | 8.18 (1.84)* | -1.14 (1.87)* | -1.23 (1.11) | 3.23 (1.27) | 3.56 (1.37) |

Notes: The absolute values of t-ratios *, ** shows coefficient estimates are at 10% and 5% levels.

1. Implement measures to stabilize exchange rates: Given the negative impact of exchange rate volatility on exports and imports in certain countries, policymakers should consider implementing measures to stabilize exchange rates. This could involve interventions in foreign exchange markets, such as the use of monetary policy tools or interventions to manage currency fluctuations and reduce volatility.

2. Enhance export competitiveness: In countries where increased exchange rate volatility has a negative impact on exports, policymakers should focus on enhancing export competitiveness. This could involve measures such as providing export subsidies, promoting diversification of export products, improving trade infrastructure, and investing in research and development to foster innovation and enhance the quality of exports.

3. Foster currency stability for import planning: Countries that experience negative long-term impacts on imports due to exchange rate volatility should aim to foster currency stability. This can be achieved through prudent monetary and fiscal policies that promote macroeconomic stability and reduce fluctuations in exchange rates. Stable currencies provide a conducive environment for import planning and reduce uncertainty for businesses.

4. Harmonize trade rules and regulations: The study highlights the influence of trade rules and regulations, such as tariffs and coherence levels, on the outcomes of exchange rate volatility on trade flows. Policymakers should work towards harmonizing trade rules and regulations to reduce barriers to trade and create a level playing field for businesses across countries. This can enhance the overall effectiveness of policies aimed at promoting trade and economic growth.

Overall, the study suggests the importance of targeted policy interventions to address the specific impacts of exchange rate volatility on trade flows in each country. By implementing measures to stabilize exchange rates, enhance export competitiveness, foster currency stability for imports, develop risk management strategies, harmonize trade rules, and continually monitor the situation, policymakers can mitigate the negative effects of exchange rate volatility and promote sustainable economic development.

## Supporting information

**S1 Appendix. Data definition and sources [46, 47].**
(DOCX)

## Author Contributions

**Conceptualization:** Umar Nawaz Kayani, Ahmet Faruk Aysan, Azeem Gul, Syed Arslan Haider, Sareer Ahmad.

**Data curation:** Umar Nawaz Kayani, Azeem Gul, Syed Arslan Haider, Sareer Ahmad.

**Formal analysis:** Umar Nawaz Kayani, Azeem Gul, Syed Arslan Haider, Sareer Ahmad.

**Funding acquisition:** Umar Nawaz Kayani, Ahmet Faruk Aysan, Azeem Gul, Syed Arslan Haider, Sareer Ahmad.

**Investigation:** Umar Nawaz Kayani, Ahmet Faruk Aysan, Azeem Gul, Syed Arslan Haider, Sareer Ahmad.

**Methodology:** Umar Nawaz Kayani, Ahmet Faruk Aysan, Azeem Gul, Syed Arslan Haider, Sareer Ahmad.

**Project administration:** Umar Nawaz Kayani, Ahmet Faruk Aysan, Azeem Gul, Syed Arslan Haider, Sareer Ahmad.

**Resources:** Umar Nawaz Kayani, Ahmet Faruk Aysan, Azeem Gul, Syed Arslan Haider, Sareer Ahmad.

**Software:** Umar Nawaz Kayani, Azeem Gul, Syed Arslan Haider, Sareer Ahmad.

**Supervision:** Umar Nawaz Kayani, Ahmet Faruk Aysan, Azeem Gul, Syed Arslan Haider, Sareer Ahmad.

**Validation:** Umar Nawaz Kayani, Ahmet Faruk Aysan, Azeem Gul, Syed Arslan Haider, Sareer Ahmad.

**Visualization:** Umar Nawaz Kayani, Ahmet Faruk Aysan, Azeem Gul, Syed Arslan Haider, Sareer Ahmad.

**Writing – original draft:** Umar Nawaz Kayani, Azeem Gul, Syed Arslan Haider, Sareer Ahmad.

**Writing – review & editing:** Ahmet Faruk Aysan, Azeem Gul, Syed Arslan Haider, Sareer Ahmad.

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
