## [Decision Letter · Decision Letter 0]

15 May 2023

PONE-D-23-08701Unpacking the Asymmetric Impact of Exchange Rate Volatility on Trade Flows: A Study of Selected Developed and Developing Asian EconomiesPLOS ONE

Dear Dr. Aysan,

Thank you for submitting your manuscript to PLOS ONE. After careful consideration, we feel that it has merit but does not fully meet PLOS ONE’s publication criteria as it currently stands. Therefore, we invite you to submit a revised version of the manuscript that addresses the points raised during the review process.

Major Revisions 

We look forward to receiving your revised manuscript.

Kind regards,

Faisal Abbas, PhD

Academic Editor

PLOS ONE

Journal Requirements:

When submitting your revision, we need you to address these additional requirements.2.

2. Please ensure that you include a title page within your main document. You should list all authors and all affiliations as per our author instructions and clearly indicate the corresponding author.

Additional Editor Comments (if provided):

Major revisions are required.

Reviewers' comments:

Reviewer's Responses to Questions

**Comments to the Author**

1. Is the manuscript technically sound, and do the data support the conclusions?

Reviewer #1: Yes

Reviewer #2: Partly

2. Has the statistical analysis been performed appropriately and rigorously? 

Reviewer #1: Yes

Reviewer #2: I Don't Know

3. Have the authors made all data underlying the findings in their manuscript fully available?

Reviewer #1: Yes

Reviewer #2: No

4. Is the manuscript presented in an intelligible fashion and written in standard English?

Reviewer #1: Yes

Reviewer #2: No

5. Review Comments to the Author

Reviewer #1: The choice of topic is interesting and good written, however, the following suggestions will make the article better and worth readable.

1. The abstract should be rewritten by adding methodology and policy implications emerging from this study.

2. The introduction section should be narrating the significance of the topic and rationale of the study.

3. The literature review section is weak and needs to be added relevant and recent literature.

4. The methodological section is good written, however, theoretical framework and choice of the variables in the model are missing. For this purpose following article will serve the purpose to some extent:

YASMEEN, R., & HAFEEZ, M. (2018). Trade balance and terms of trade relationship: Evidence from Pakistan. Pakistan Journal of Applied Economics, 28(2), 173-188.

5.The study uses GARCH model for volatility measurement. There are many other methods why only the GARCH is preferred on others, Justify.

6. The choice of the sample countries should be based on some logic. It should be justified.

7. Mostly, the results of the coefficients estimates for different countries are different. The differences should be explained logically.

Reviewer #2: 1. The manuscript lacks coherence, there is large amount of irrelevant literature in introduction section that is not inline with objectives.

2. The equations in methodology section are not clear because there are no proper subscripts.

3. Authors have not provided any justification for using the selected developing and developed countries.

4. The findings and conclusions are not inline with the objectives given in introduction section.

The key objectives were to (i) evaluate the response of exchange rate on export

and import in selected Asian countries, (ii) change in trade flows, (iii) encourage policy

coordination, and (iv) improve connectivity. However, these objectives can not be achieved with the given methodology except (i). Rest of the objectives are not even discussed in results and conclusion section.

6. PLOS authors have the option to publish the peer review history of their article (what does this mean?). If published, this will include your full peer review and any attached files.

Reviewer #1: No

Reviewer #2: No

---

## [Author Response · Author response to Decision Letter 0]

1 Jul 2023

Unpacking the Asymmetric Impact of Exchange Rate Volatility on Trade Flows: A Study of Selected Developed and Developing Asian Economies

PONE-D-23-08701

Response to the Editor / Referees 

Dear Editor/Referee,

Thank you very much for reviewing our manuscript, encouraging comments, constructive feedback, and valuable suggestions. We have worked diligently to address each of your remarks and suggestion and are resubmitting a substantially improved version of the paper. We include below your comments in italics, and our responses follow. Please also see the changes in yellow in the revised version of the paper. 

Referee 1

1. The abstract should be rewritten by adding methodology and policy implications emerging from this study. 

Response: Thank you for your valuable comments. We have rewritten the abstract. In the abstract, we have incorporated the objective of the study, the methodology applied, major findings, and implications of our study.

2. The introduction section should be narrating the significance of the topic and rationale of the study

Response: Thank you very much for identifying one important weakness of our initial submission. Now, we have incorporated the significance of the study and added new citations from the PLOS ONE journals.

3. The literature review section is weak and needs to be added to relevant and recent literature.

Response: Thank you for your valuable comments. We have revised and added some new citations in the literature section.

4. The methodological section is good written, however, theoretical framework and choice of the variables in the model are missing. For this purpose, following article will serve the purpose to some extent:

Response: Thank you very much for this comment which helps us to revisit our initial understanding. In our revised submission, we have included a revised methodology and explained the Model in detail.

5. The study uses GARCH model for volatility measurement. There are many other methods why only the GARCH is preferred on others, Justify.

Response: Thank you very much for this comment which helps us to revise our initial understanding. In our revised submission, we have included the logic of why we used GARCH model and why we are not using other methods in this study. 

6. The choice of the sample countries should be based on some logic. It should be justified.

Response: Thank you very much. Now, we have incorporated justifications behind the selection of the Asian developed and developing countries (page 7). 

7. Mostly, the results of the coefficients estimate for different countries are different. The differences should be explained logically.

Response: Thank you very much. We have revised the equations that have wrong coefficient signs We also revised the interpretations of the results and add why we used asymmetry. 

Referee 2

1. The manuscript lacks coherence, there is large amount of irrelevant literature in introduction section that is not in line with objectives.

Response: Thank you very much for encouraging comments regarding our work. We are happy and delighted to receive such appreciation from learned reviewer. We have revised our objective of the study, and revised introduction part and also add some new citations which is coherent with our objectives.

2. The equations in methodology section are not clear because there are no proper subscripts.

Response: Thank you very much for identifying one of the major weaknesses of our initial submission. We have deployed our best effort to improve the methodology section. Now we have changed the equations and corrected the values of the coefficients.

3. Authors have not provided any justification for using the selected developing and developed countries.

Response: Thank you very much for this comment which helps us to revisit our initial understanding. In our revised submission, we have included the justification for why we have selected Asian developing and developed nations in this study (page 7).

4. The findings and conclusions are not in line with the objectives given in the introduction section.

Response: Thank you for this helpful comment. We have revised our study objective and rewrite the conclusion of the study. And on the bases of our findings, we also add some policy recommendations.

In conclusion, we are thankful to you for your constructive feedback on our submission, which has improved the paper significantly. We have tried our best to address the comments in our revised version and believe that the paper has been changed substantially. We will be waiting for your valuable comments on our revised submission, please.

---

## [Decision Letter · Decision Letter 1]

19 Jul 2023

PONE-D-23-08701R1Unpacking the Asymmetric Impact of Exchange Rate Volatility on Trade Flows: A Study of Selected Developed and Developing Asian EconomiesPLOS ONE

Dear Dr. Aysan,

Thank you for submitting your manuscript to PLOS ONE. After careful consideration, we feel that it has merit but does not fully meet PLOS ONE’s publication criteria as it currently stands. Therefore, we invite you to submit a revised version of the manuscript that addresses the points raised during the review process.

Minor Revision

We look forward to receiving your revised manuscript.

Kind regards,

Faisal Abbas, PhD

Academic Editor

PLOS ONE

Journal Requirements:

Additional Editor Comments:

Minor Revision.

Reviewers' comments:

Reviewer's Responses to Questions

**Comments to the Author**

1. If the authors have adequately addressed your comments raised in a previous round of review and you feel that this manuscript is now acceptable for publication, you may indicate that here to bypass the “Comments to the Author” section, enter your conflict of interest statement in the “Confidential to Editor” section, and submit your "Accept" recommendation.

Reviewer #1: All comments have been addressed

Reviewer #2: (No Response)

2. Is the manuscript technically sound, and do the data support the conclusions?

Reviewer #1: Yes

Reviewer #2: Partly

3. Has the statistical analysis been performed appropriately and rigorously? 

Reviewer #1: Yes

Reviewer #2: (No Response)

4. Have the authors made all data underlying the findings in their manuscript fully available?

Reviewer #1: Yes

Reviewer #2: (No Response)

5. Is the manuscript presented in an intelligible fashion and written in standard English?

Reviewer #1: Yes

Reviewer #2: Yes

6. Review Comments to the Author

Reviewer #1: The article has discussed the Impact of Exchange Rate Volatility on Trade Flows of Developed and Developing Asian Economies. It is in better shape now and can be published.

Reviewer #2: There are still some language issues. I'm highlighting some of these issue here. It is suggested that you review the whole manuscript from an expert.

1. There must be some typo at second last line of page at where you have written "e-i." if its not a type please explain this term.

2.On page 9 first paragraph, you are first time using word "GARCH", please give full form when you are first time using it. Similarly for other abbreviations use full form first time.

3. In Table 1 value in 8th column are missing. Some other tables also have same issue, last column is empty.

4. There are typos in last paragraph of page 9, where you have written " We anticipate the estimate of B1 and 1 to be positive....." Similarly, you have written " We anticipate the estimate of 2 to be negative". Its not clear whether you referring to equation or coefficient.

7. PLOS authors have the option to publish the peer review history of their article (what does this mean?). If published, this will include your full peer review and any attached files.

Reviewer #1: No

Reviewer #2: No

---

## [Author Response · Author response to Decision Letter 1]

12 Aug 2023

PONE-D-23-08701R1

Unpacking the Asymmetric Impact of Exchange Rate Volatility on Trade Flows: A Study of Selected Developed and Developing Asian Economies

PLOS ONE

Thank you for submitting your manuscript to PLOS ONE. After careful consideration, we feel that it has merit but does not fully meet PLOS ONE’s publication criteria as it currently stands. Therefore, we invite you to submit a revised version of the manuscript that addresses the points raised during the review process.

Minor Revision

We look forward to receiving your revised manuscript. 

Kind regards,

Faisal Abbas, PhD

Academic Editor

PLOS ONE

Journal Requirements:

Additional Editor Comments:

Minor Revision.

Reviewers' comments:

Reviewer's Responses to Questions 

Comments to the Author

1. If the authors have adequately addressed your comments raised in a previous round of review and you feel that this manuscript is now acceptable for publication, you may indicate that here to bypass the “Comments to the Author” section, enter your conflict of interest statement in the “Confidential to Editor” section, and submit your "Accept" recommendation.

Reviewer #1: All comments have been addressed

Reviewer #2: (No Response)

2. Is the manuscript technically sound, and do the data support the conclusions?

Reviewer #1: Yes

Reviewer #2: Partly

3. Has the statistical analysis been performed appropriately and rigorously? 

Reviewer #1: Yes

Reviewer #2: (No Response)

4. Have the authors made all data underlying the findings in their manuscript fully available?

Reviewer #1: Yes

Reviewer #2: (No Response)

5. Is the manuscript presented in an intelligible fashion and written in standard English?

Reviewer #1: Yes

Reviewer #2: Yes

6. Review Comments to the Author

Reviewer #1: The article has discussed the Impact of Exchange Rate Volatility on Trade Flows of Developed and Developing Asian Economies. It is in better shape now and can be published.

Reviewer #2: There are still some language issues. I'm highlighting some of these issue here. It is suggested that you review the whole manuscript from an expert.

1. There must be some typo at second last line of page at where you have written "e-i." if its not a type please explain this term.

Response:

2.On page 9 first paragraph, you are first time using word "GARCH", please give full form when you are first time using it. Similarly for other abbreviations use full form first time.

Response: Thank you so much. The term GARCH stand for Generalized Autoregressive Conditional Heteroskedasticity (GARCH) and we incorporate the comment in page 9.

3. In Table 1 value in 8th column are missing. Some other tables also have same issue, last column is empty.

Response: Thank you for your comment. Actually, the logic of missing values is that we run the data through E-Views software and we have selected automatic selection in E-views due to this the software takes the lag values according to the model and we have not selected itself that’s the reason some columns values are missing.

4. There are typos in last paragraph of page 9, where you have written " We anticipate the estimate of B1 and 1 to be positive....." Similarly, you have written " We anticipate the estimate of 2 to be negative". Its not clear whether you referring to equation or coefficient

Response: Thank you for your comment. We have checked the manuscript and revised the manuscript. The word/ typo errors have been corrected in the manuscript.

7. PLOS authors have the option to publish the peer review history of their article (what does this mean?). If published, this will include your full peer review and any attached files.

Do you want your identity to be public for this peer review? For information about this choice, including consent withdrawal, please see our Privacy Policy.

Reviewer #1: No

Reviewer #2: No

---

## [Decision Letter · Decision Letter 2]

25 Aug 2023

Unpacking the Asymmetric Impact of Exchange Rate Volatility on Trade Flows: A Study of Selected Developed and Developing Asian Economies

PONE-D-23-08701R2

Dear Dr. Aysan,

We’re pleased to inform you that your manuscript has been judged scientifically suitable for publication and will be formally accepted for publication once it meets all outstanding technical requirements.

Kind regards,

Faisal Abbas, PhD

Academic Editor

PLOS ONE

Additional Editor Comments (optional):

Accept

Reviewers' comments:

Reviewer's Responses to Questions

**Comments to the Author**

1. If the authors have adequately addressed your comments raised in a previous round of review and you feel that this manuscript is now acceptable for publication, you may indicate that here to bypass the “Comments to the Author” section, enter your conflict of interest statement in the “Confidential to Editor” section, and submit your "Accept" recommendation.

Reviewer #2: All comments have been addressed

2. Is the manuscript technically sound, and do the data support the conclusions?

Reviewer #2: Yes

3. Has the statistical analysis been performed appropriately and rigorously? 

Reviewer #2: Yes

4. Have the authors made all data underlying the findings in their manuscript fully available?

Reviewer #2: Yes

5. Is the manuscript presented in an intelligible fashion and written in standard English?

Reviewer #2: Yes

6. Review Comments to the Author

Reviewer #2: Authors have incorporated the comments and provided justifications where required. The manuscript has improved a lot and have clear and logical flow of arguments.

7. PLOS authors have the option to publish the peer review history of their article (what does this mean?). If published, this will include your full peer review and any attached files.

Reviewer #2: No

---

## [Editor Report · Acceptance letter]

2 Oct 2023

PONE-D-23-08701R2 

Unpacking the Asymmetric Impact of Exchange Rate Volatility on Trade Flows: A Study of Selected Developed and Developing Asian Economies 

Dear Dr. Aysan:

I'm pleased to inform you that your manuscript has been deemed suitable for publication in PLOS ONE. Congratulations! Your manuscript is now with our production department. 

Kind regards, 

on behalf of

Dr. Faisal Abbas 

Academic Editor

PLOS ONE